# Short-Term Exposure to Nanoplastics Does Not Affect Bisphenol A Embryotoxicity to Marine Ascidian *Ciona robusta*

**DOI:** 10.3390/biom12111661

**Published:** 2022-11-08

**Authors:** Emma Ferrari, Maria Concetta Eliso, Arianna Bellingeri, Ilaria Corsi, Antonietta Spagnuolo

**Affiliations:** 1Department of Physical, Earth and Environmental Sciences, University of Siena, 53100 Siena, Italy; 2Department of Biology and Evolution of Marine Organisms, Stazione Zoologica Anton Dohrn, 80121 Naples, Italy; 3Department of Sciences, Roma Tre University, 00146 Rome, Italy

**Keywords:** polystyrene nanoparticles, bisphenol A, ecotoxicity, Mediterranean Sea, ascidian, embryo development

## Abstract

Plastic pollution is recognized as a global environmental threat and concern is increasing regarding the potential interactions of the smallest fragments, nanoplastics (1 µm), with either physical and chemical entities encountered in the natural environment, including toxic pollutants. The smallest size of nanoplastics (<100 nm) rebounds to their safety associated with remarkable biological, chemical and physical reactivity that allow them to interact with cellular machinery by crossing biological barriers and causing damage to living beings. Recent findings on nanoplastic occurrence in marine coastal waters, including the Mediterranean Sea, leave open the question on their ability to act as a vector of other contaminants of emerging concerns (CECs) concomitantly released by wastewater treatment plants and reaching marine coastal waters. Here, we assess for the first time the role of non-functionalized polystyrene nanoparticles (PS NPs, 20 nm) as a proxy for nanoplastics (1 and 10 µg/mL) alone and in combination with bisphenol A (BPA) (4.5 and 10 µM) on *Ciona robusta* embryos (22 h post fertilization, hpf) by looking at embryotoxicity through phenotypic alterations. We confirmed the ability of BPA to impact ascidian *C. robusta* embryo development, by affecting sensory organs pigmentation, either alone and in combination with PS NPs. Our findings suggest that no interactions are taking place between PS NPs and BPA in filtered sea water (FSW) probably due to the high ionic strength of seawater able to trigger the sorption surface properties of PS NPs. Further studies are needed to elucidate such peculiarities and define the risk posed by combined exposure to BPA and PS NPs in marine coastal waters.

## 1. Introduction

Plastic pollution is a global problem affecting oceans around the world and poses a serious threat in marine coastal areas where they encounter/interact with other contaminants released by human activities [1,2,3]. The smallest fragments, named nanoplastics (<1 µm), rebound to their environmental safety associated with unique biological, chemical and physical reactivity that allow them to interact with living beings by crossing biological barriers and eventually causing damage to marine organisms [4,5]. Nanoplastics usually refer to polymers with various sizes, colors and shapes, thus encompassing the sub-micron and nanometric fraction [6]. A heavy load of nanoplastics is expected in marine coastal areas due to waste products released from their production, usage and disposal (primary sources), as well as the environmental breakdown of larger bulk materials unintentionally produced due to mechanical processes, weathering agents and biota (secondary sources) [7,8,9,10]. Synthetic nano-polymers are used in industrial and commercial products, including biosensors, photonics [11], cosmetics [12,13], food nanocomposites [14] and pharmaceutical drug nanocarriers [15]. Modeled predicted environmental concentrations (PECs) for nanoplastics in surface waters (<20 µg/L) are expected to increase in the near future, particularly in hot spot areas such as marine coastal areas [16,17]. 

Recent findings on nanoplastic occurrence in Mediterranean coastal waters leave open the question on their ability to act as vectors of other contaminants of emerging concerns (CECs), concomitantly released by wastewater treatment plants, and of additives and other known toxic chemicals associated with polymer synthesis [18,19]. While the interactions of larger plastics as microplastics with other organic and inorganic pollutants has been studied over the last decade, very little is known on such interactions with nanoplastics and in particular in marine waters [20,21]. 

The information obtained from studies with microplastics may not necessarily be transferred to nanoplastics due to their unique properties at the nano-level, which include high surface reactivity and the ability to carry adsorbed chemicals into organisms/cells. In addition, natural seawater might modify NP properties by affecting particle size (i.e., agglomeration) and surface charges (i.e., environmental corona) so that the adsorption property and the potential *trojan horse* effect towards other chemicals can be limited [22,23]. Our previous findings on polystyrene nanoparticle (PS NPs) incubation in natural sea water (NSW) revealed significant changes in ζ-potential and ζ-average with the formation of an environmental corona which conferred new properties to the surface of the PS NPs [5]. A protein corona formation upon incubation with biological fluid of a marine species further confirmed how the surface of PS NPs can react with the external water milieu [24,25]. Such changes might significantly affect any chemical interaction with other compounds present in the NSW media, starting from their adsorption properties [26]. Bisphenol A (BPA) is a well known endocrine disruptor used in the polycarbonate plastic and epoxy resins [27,28]. BPA is ubiquitous in the environment and has recently been banned due to its ability to migrate into the aqueous phase. It can enter the body through breathing, food, and skin, thus inducing multi-organ toxicity in humans and animals, causing reproductive, immunity and neurological problems [29]. While BPA interactions with micro and nanoplastics have been investigated in freshwater media, showing that both bioaccumulation and toxicity were exacerbated by combined exposure, interactions and ecotoxicity on marine waters and biota have been overlooked. As a model of ecotoxicological studies, in recent years, the ascidian *Ciona robusta* is increasingly being used; it is an invertebrate *Urochordate* which shows the peculiarity of being a simplified chordate developmental model at the larval stage ([30] and references therein) and is an abundant component in marine fauna. Previous studies have analyzed BPA effects on *Ciona* embryonic development, demonstrating its sensitivity to BPA [31,32]. Here, we investigated for the first time the potential interaction of non-functionalized PS NPs (20 nm), as a proxy for environmental nanoplastics, with BPA in natural seawater media at environmentally realistic concentrations. We evaluated the effects of of PS NPs (1 and 10 µg/mL) alone and in combination with BPA (4.5 and 10 µM) on *Ciona robusta* embryogenesis, by looking at embryotoxicity and phenotypic alterations, to inspect any potential or not synergistic effect when both are present in marine environment during this critical period of life.

## 2. Materials and Methods

### 2.1. Organism Tested

Adult specimens of ascidian *C. robusta* were collected in Fusaro Lake (Italy) by local fishermen between March 2022 and June 2022 and immediately shipped in cool boxes to the aquarium facility of the Zoological Station Anton Dohrn of Naples (Italy). An acclimation of 7 days was performed in flow-through circulating aquarium using NSW (18 ± 1 °C; salinity 40 ± 1‰, dissolved O_2_ 7 mg/L; pH 8.1) under constant aeration and continuous light to stimulate gametes maturation but to avoid spawning [33]. Individuals were fed *ad libitum* every 48 h with algal mix (Shellfish Diet 1800^®^). Gametes were obtained from each individual by dissecting the gonoducts with a scalpel. To avoid the self-fertilization, oocytes and sperms were collected by distinct individuals. Fertilization was performed by adding diluted sperm (1:100 in Filtered Sea Water (FSW)) to the egg suspension and after 10 min of incubation on a rotating shaker, the fertilized eggs were transferred to tissue culture plates [33].

### 2.2. BPA and PS NPs Preparation and Characterization

Non-functionalized 20 nm PS NPs were purchased from Bangs Laboratories Inc. (Fishers, IN, USA) and received as stock suspensions (100 µg/mL) in deionised water. After a brief bath sonication (15 min, 70% power), PS NPs working suspensions (10 mg/mL and 1 mg/mL) were prepared in MilliQ (mQW) and stored in sterile vials at 4 °C until use. Bisphenol A (BPA) was purchased from Sigma-Aldrich (St. Louis, MO, USA) (CAS Nr. 80-05-7), dissolved in 0.1 M dimethyl sulfoxide (DMSO Sigma-Aldrich CAS Nr. 67-68-5) and then further diluted to 1 mM in FSW. PS NPs stock and working suspensions in mQW and FSW (0.45 µm, temperature 18 ± 1 °C; dissolved O_2_ 7 mg/L; pH 8.1) were measured by DLS for ζ-average, polydispersity index (PDI) and ζ-potential (50 µg/mL) using a Zetasizer Nano ZS90 (Malvern, Malvern, UK) equipped with the Zetasizer Nano Series software (Ver. 7.02). PS NPs behavior variations between Time 0 and Time 24 (24 h incubation) alone and in combination with BPA in FSW at the highest concentration tested (10 µM), following the experimental design of embryotoxicity assay, were also investigated. The PS NPs suspension was diluted in FSW to obtain the two tested concentrations of 0.1 µg/mL and 1 µg/mL and used without further sonication according to the protocol reported in Eliso et al. [33]. Two controls, one in FSW and another in FSW with DMSO (0.01%), were also established. BPA treatment concentrations were chosen following trial experiments and literature searches: 4.5 µM was chosen as the EC_50_ for *C. robusta* larvae for specific malformations given by BPA, i.e., the reduced pigmentation of the sensory organs and 10 µM as the highest concentration at which morphological effects were observed in most of the individuals tested [34].

### 2.3. Embryotoxicity Assays

The experimental design based on a time-dependent combined exposure was run as follows: (i) PS NPs and BPA were tested alone and combined by adding working solution and particle suspensions in FSW and immediately tested with ascidian embryos (Time 0); (ii) PS NPs and BPA were pre-incubated in FSW alone, combined for 24 h and then tested with the ascidian embryos (Time 24). BPA (4.5 µM, 10 µM) was mixed in Falcon—50 mL (Conical Centrifuge Tubes) with PS NPs (1 µg/mL, 0.1 µg/mL) in FSW (20 mL) and left at room temperature at 25 ± 2 °C for 24 h using a gentle rotating shaker (150 rpm), according to the protocol described by Chen et al. [35]. The embryotoxicity assay was run according to the method of Lambert et al., [36] and Bellas et al. [37]: one hour post-fertilization (hpf), 60 embryos (two-cell stage) were placed in 6-well plates and respectively exposed to FSW, FSW + 0.01% DMSO, PS NPs and BPA alone (4.5 and 10 µM) and in combination either without incubation (Time 0) and incubated for 24 h as follows: PS NPs 0.1–1 µg/mL with BPA 4.5 and 10 µM, respectively, in combination at the lowest and highest concentrations. The impact of PS NPs and BPA alone and in combination was evaluated as the percentage of normal hatched larvae and morphological alterations at 22 hpf compared to controls (FSW and 0.01% DMSO), as described below. Larvae were first fixed in 4% paraformaldehyde and then washed twice in 1X PBS. Several endpoints, such as lethality, hatching, growth, morphological alterations and larvae viability were examined. Each larva was assessed as normal according to the Tunicate Anatomical and Developmental Ontology (https://www.bpni.bio.keio.ac.jp/tunicanato/3.0/developmental_table.html [38], accessed on 30 September 2022). Larval phenotypes were observed by using the stereomicroscope Zeiss Axio Imager M1 and classified as follows: normal, malformed, reduced pigmentation and not developed. The assay was considered valid when controls showed a percentage of normal hatched larvae ≥ 70% at 22 hpf.

### 2.4. Statistical Analysis

Data are shown as mean ± standard deviation (SD). Statistical analysis was performed using GraphPad Prism software (Version 8.0.1, San Diego, CA, USA) using 2-way ANOVA with a Tukey multiple comparisons test. Significance is indicated in the figures as * *p* < 0.05.

## 3. Results 

### 3.1. Behavior of PS NPs in MilliQ and Filtered Natural Seawater (FSW) 

PS NPs showed a ζ-average (nominal hydrodynamic diameter) of 22.8 ± 0.3 nm in mQW with a relatively wide distribution of sizes (PDI value 0.35 ± 0.4), and a negative surface charge (−51 ± 5 mV) (Table 1). An increase in ζ-potential values and the formation of agglomerates occur when PS NPs are dispersed in FSW at time 0 h (1589 ± 139 nm) and more pronounced in the presence of BPA (2583 ± 198 nm). 24 h incubation in FSW, with or without BPA, increases agglomerates size (>3000 nm) and results in a highly polydisperse scenario (PDI 0.5–0.7) on which either ζ-average and ζ-potential values are too wide to be considered descriptive of PS NPs batch analyzed.

### 3.2. Embryotoxicity Assays of PS NPs and BPA Alone 

PS NPs did not cause any significant effect on the development of *C. robusta* embryos at both concentrations tested (0.1 and 1 µg/mL), with only a slight increase in the % of undeveloped embryos compared to the controls (Figure 1; Table 2). Conversely, larvae exposed to BPA, as previously reported [34] showed two types of malformations: (i) a reduced pigmentation of both otolith and ocellus (62.46% at 4.5 µm and 45.38% at 10 µM) (Table 2, Figure 1, mild malformations), the two pigmented sensory organs present in the brain vesicle (Figure 2—C’ compared to the controls A,A’); (ii) a general compromised development with a shorter, kinked and disorganized tail and an altered trunk morphology (Table 2, Figure 1, general malformations) (Figure 2D,E). Furthermore, at the highest concentration of BPA, there was a significant percentage of undeveloped embryos, compared to BPA 4.5 µM (Table 2).

### 3.3. Embryotoxicity Assays with PS NPs and BPA Combined (T0)

Figure 3 shows the results of the combined exposure without pre-incubation in FSW (T0). The percentage of normal larvae was significantly decreased at all tested conditions compared to values of controls. The percentage of low pigmented larvae (mild malformations) resulted very high in all tested conditions (Figure 3), including the combination at the highest PS NPs and BPA concentrations (PS NPs 1 µg/mL + BPA 10 µM), which also showed the high % of undeveloped embryos (55.47%, Table 3). By comparing the % of malformed larvae, between single and combined exposure, no differences were observed, with trends very similar to those observed for BPA alone. Similarly, the effects on pigmentation were visually the same upon exposure to BPA alone and in combination with PS NPs (Figure 2).

### 3.4. Embryotoxicity Assays with PS NPs and BPA Combined (T24) 

Figure 4 shows the effects of combined exposure upon 24 h of incubation of BPA with PS NPs in FSW. Again, the percentage of normal larvae was significantly reduced at all concentrations of combined BPA and PS NPs, compared to controls (Table 4). On the other hand, the percentages of normal larvae resulted higher compared to the combined exposure without prior incubation (T0) (see Figure 3), suggesting a degree of interaction between PS NPs and BPA after 24 h incubation in FSW. Only in the group exposed to the lowest PS NPs (0.1 µg/mL) and the highest BPA (10 µM) concentrations, no normal larvae were found, resulting in the highest % of undeveloped larvae (78.39%). Conversely, a small % of larvae showed reduced pigmentation (14.05%), while under the other experimental conditions, the effects on pigmentation were in line with the other two experiments and were confirmed in all of them above 50% (Table 4).

## 4. Discussion

The present study aims to fill the current knowledge gap on the potential interaction of non-functionalized PS NPs with BPA in NSW at environmentally realistic concentrations. Embryotoxicity and phenotypic alterations observed in *C. robusta* larvae upon exposure to single and combined PS NPs and BPA suggest that interaction is not taking place probably due to the high ionic strength of NSW which affects PS NPs surface charges. BPA has been confirmed to affect *C. robusta* larvae pigmentation in agreement with previous studies on freshwater species [34]. BPA is known as an endocrine disruptor targeting several vertebrate membrane receptors (ER, AhR, PPAR, PXR, TR) and mainly nuclear receptors in invertebrates, thus being responsible for neurodevelopmental effects [31]. At environmentally realistic concentrations, which can vary significantly in the range 0.1–75 µM, BPA is reported to be toxic to aquatic species by mainly affecting embryo development either in vertebrates or invertebrates [39,40]. A dose-dependent embriotoxicity has been reported for BPA in other ascidian embryos as *Phallusia mammillata* (EC_50_: 11.8 µM; LC_50_: 21 µM) and at lower µM concentrations an impairment in pigmented cells differentiation, by inhibiting otolith movement within the sensory vesicle, has been described [32,34,41]. Similarly, BPA is able to cause less pigmentation in zebrafish embryos (5.0 mg/L) by downregulating the melanin synthases and then reducing the melanin content. The role of BPA as a ligand of zebrafish tyrosinase (Tyr) family proteins has been hypothesized, thus causing skin pigmentation interference [42]. Here, we confirmed the ability of BPA to affect ascidian *C. robusta* embryo development even in combination with PS NPs. Further confirmation of the effect of BPA alone on *C. robusta* larvae pigmentation (4.5 and 10 µM) is provided, which seems to be not affected by PS NPs exposure.

Combined exposure to PS NPs in FSW at either concentrations tested (0.1 and 1 µg/mL) did not affect BPA effects on *C. robusta* larvae both at time 0 and after 24 h incubation in FSW, for which no differences with time have been observed in either general malformed or mild malformations. We, therefore, based on our findings, hypothesize that no interaction is taking place between BPA and PS NPs in the seawater media. Conversely, co-exposure of PS NPs (50 nm) and BPA (1 µg/L and 1 mg/L) in artificial freshwater media has been reported to significantly increase (2.2/2.6-fold) BPA uptake in the head and viscera of adult zebrafish upon short-term exposure (3d) [35]. The absence of an interactive effect of PS NPs towards BPA in FSW in our study could be explained by DLS data which show the formation of large agglomerates (ζ-average 1589 ± 139) upon PS NPs dispersion in FSW reaching micrometric size and becoming even larger upon co-incubation with BPA (2583 ± 198) (Table 1). Such behavior is further increased upon incubation in FSW for 24 h with the formation of huge agglomerates (<3000) either alone or in the presence of BPA (<3000). Such a PS NPs agglomeration phenomenon has been previously described and occurs soon after dispersion in aqueous media characterized by high ionic strength (>38 h), basic pH, and NOM as FSW and supported by the DLVO theory and related extensions ([5] and reference within). Furthermore, a recent study [43] showed that incubation in alkaline medium and the presence of electrolytes (i.e., NaCl) prevent the adsorption of BPA on PS NPs and induce the formation of agglomerates which affect the PS NPs adsorption behaviour.

Significant changes in PS NPs ζ-potential upon dispersion in mQW, compared to FSW, have been observed in our study with a decrease in negative surface charges of PS NPs (from −51 ± 5 in mQW to −34 ± 4 at time 0 and −31 ± 9 after 24 h in FSW). Such a decrease in ζ-potential could probably affect the adsorption properties of PS NPs towards BPA. According to Chen et al. [35], BPA is able to adsorb onto PS NPs in freshwater media and the sorption equilibrium is reached after 1 day. Our hypothesis, corroborated by DLS ζ-average and ζ-potential data, is that FSW is triggering the surface properties of PS NPs by reducing negative surface charges and the BPA sorption ability. Furthermore, upon incubation in FSW for 24 h + BPA, ζ-potential is further increased (Table 1) even compared to values at T0. Larger agglomerates formation after incubation with FSW might have entrapped BPA and thus reduced its bioavailability and embryotoxicity to *Ciona* embryos. As shown in Figure 3 and Figure 4, the % of normal embryos is higher after 24 h compared to T0, and in particular at the highest PS NPs concentration of 1 µg/mL. PS NPs at lower concentration (0.1 µg/mL) might still be better dispersed in NSW and therefore less able to retain BPA while at higher concentration (1 µg/mL); the formation of large agglomerates could have increased such a retention effect and explain the observed lower % of undeveloped embryos (Table 4) (78.39 vs. 26.54) (Figure 5).

## 5. Conclusions

Here, we perform a preliminary assessment of the effects of non-functionalized PS NPs as a proxy for nanoplastics (1–10 µg/mL) alone and in combination with BPA (4.5 and 10 µM) on *C. robusta* embryos by examining embryotoxicity through phenotypic alterations. While we confirmed the ability of BPA to affect ascidian *C. robusta* embryo development, the combined exposure with PS NPs suggest that no interaction is taking place between PS NPs and BPA in sea water media. The high ionic strength of seawater affects surface charges of PS NPs, and thus the ability to adsorb BPA. Further studies are needed to thoroughly clarify PS NPs and BPA interaction in seawater; certainly, exposure media is playing a significant role in nanoplastic interaction with other CECs and consequently on their ecotoxicity.

## Figures and Tables

**Figure 1 biomolecules-12-01661-f001:**
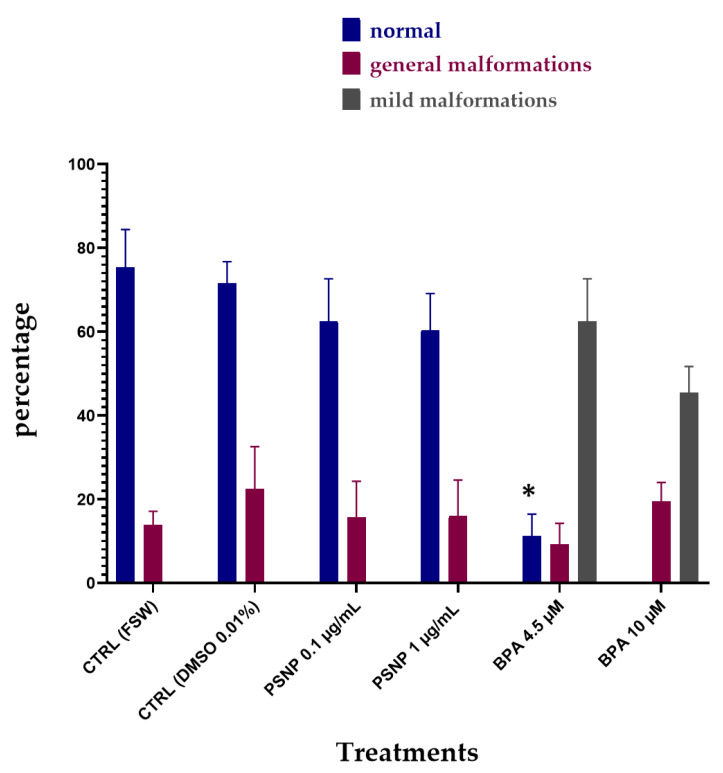
The percentage (%) of normal hatched larvae (blue), larvae with general malformations (red) and with mild malformations (reduced pigmentation of otolith and ocellus, grey) upon single exposure of 22 hpf to PS NPs (0.1 µg/L and 1 µg/L) and BPA (4.5 and 10 µM) in FSW. Bars show values expressed as mean ± Standard Deviation. * indicates values that are significantly different compared to controls (2-way ANOVA, *p* < 0.05).

**Figure 2 biomolecules-12-01661-f002:**
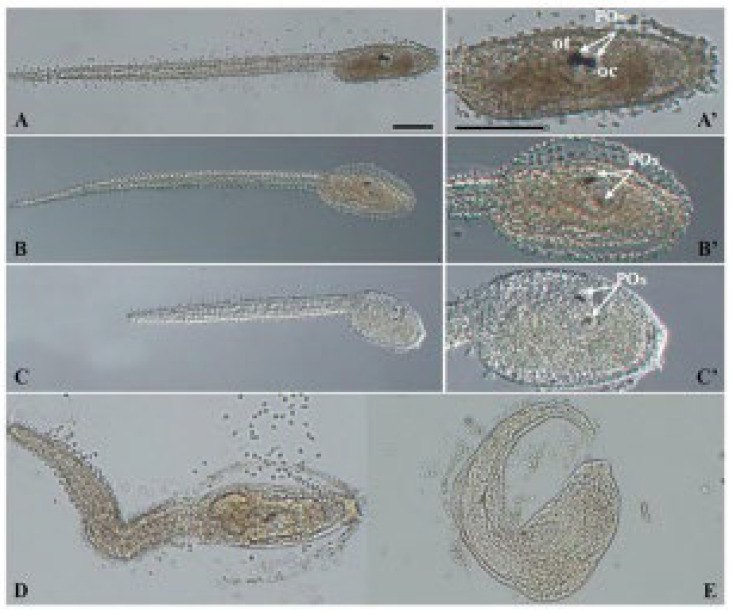
Light microscopy images of representative *C. robusta* larvae with general malformations and mild malformations (reduced pigmentation) upon exposure to BPA alone and combined with PS NPs: controls (**A**,**A’**), mild malformations (**B**,**B’**,**C**,**C’**) and general malformations (**D**,**E**). White arrows indicate otolith (ot) and ocellus (oc) which together form the pigmented organs (POs). Scale bars: 100 µm.

**Figure 3 biomolecules-12-01661-f003:**
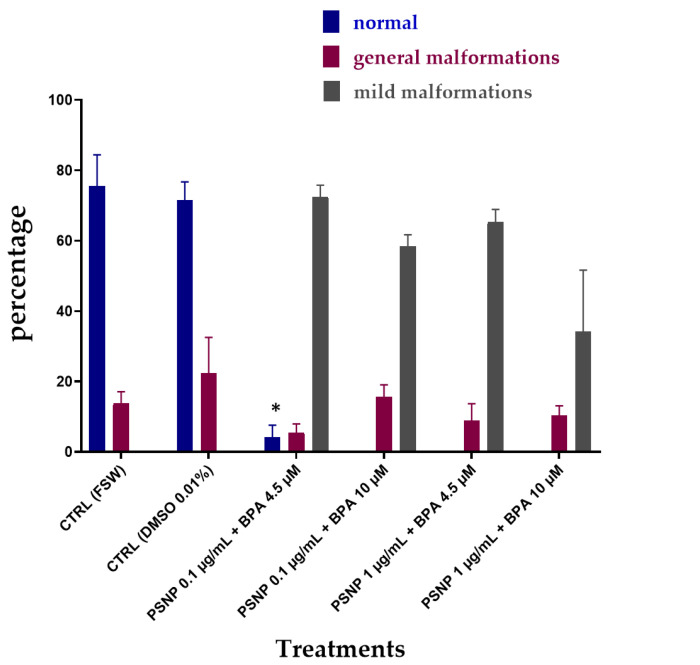
The percentage (%) of *C. robusta* larvae normally hatched with general malformations (red) and with mild malformations (grey) upon combined exposure without pre-incubation (T0) to PS NPs and BPA in FSW. Bars show mean ± Standard Deviation. * indicates values that are significantly different compared to the controls (2-way ANOVA, *p* < 0.05).

**Figure 4 biomolecules-12-01661-f004:**
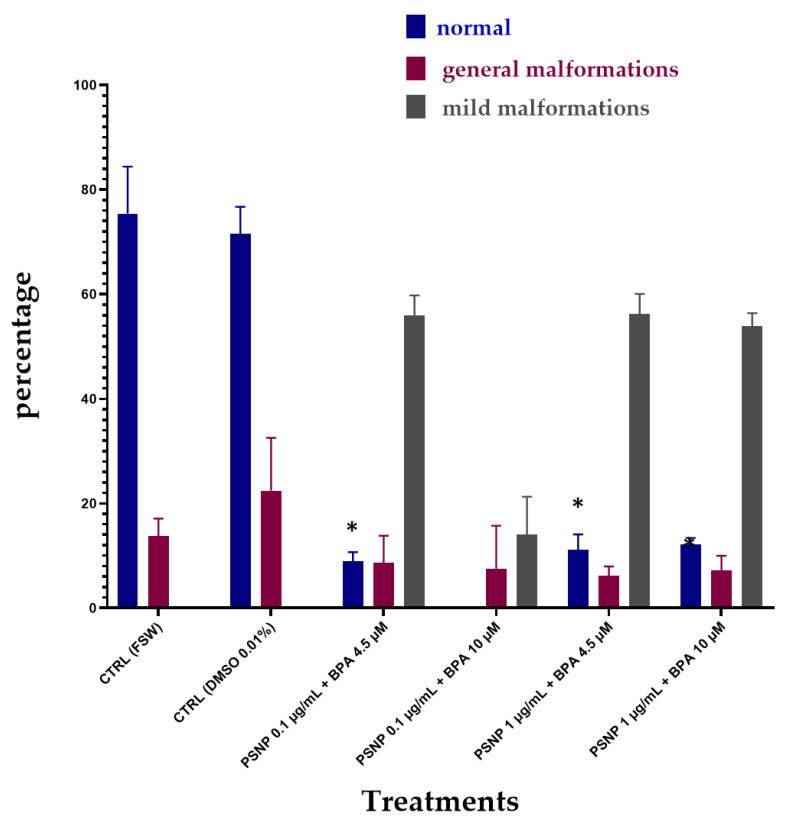
The percentage (%) of *C. robusta* normally hatched larvae, with general malformations (red) and with mild malformations (grey) upon combined exposure after 24 h of incubation (T24) in FSW to PS NPs and BPA in FSW. Bars show mean ± Standard Deviation. * indicates values that are significantly different compared to the controls (2-way ANOVA, *p* < 0.05).

**Figure 5 biomolecules-12-01661-f005:**
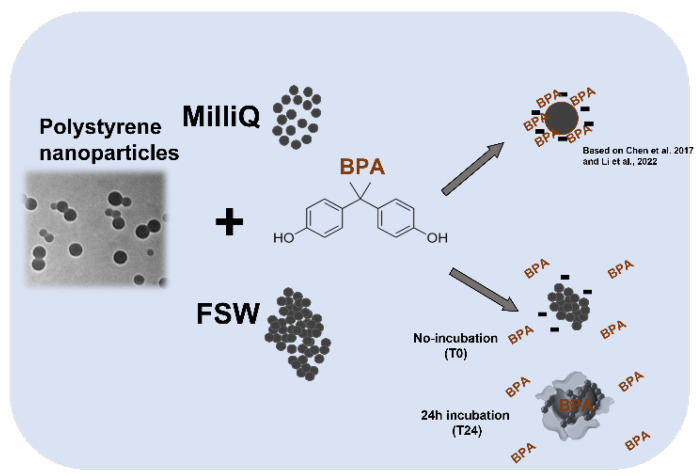
A diagram showing the possible interaction between PS NPs and BPA in MilliQ and FSW based on DLS data and biological findings. Refer to Chen et al., 2017 and Li et al., 2022 [35,43].

**Table 1 biomolecules-12-01661-t001:** ζ-average, polydispersity index (PDI) and ζ-potential measured by Dynamic Light Scattering (DLS) of non-functionalized PS NPs (50 µg/mL) in mQW, FSW and FSW with BPA (10 µM), at time 0 (0 h) and after 24 h of incubation (24 h). Values are shown as mean ± standard deviation.

	ζ-Average (nm)	PDI	ζ-Potential (mV)
mQW	22.8 ± 0.3	0.35 ± 0.04	−51 ± 5
FSW 0 h	1589 ± 139	0.35 ± 0.04	−34 ± 4
FSW + BPA (10 µM) 0 h	2583 ± 198	0.46 ± 0.1	−32 ± 4
FSW 24 h	>3000	0.76 ± 0.3	−31 ± 9
FSW + BPA (10 µM) 24 h	>3000	0.53 ± 0.4	−21 ± 14

**Table 2 biomolecules-12-01661-t002:** The percentage (%) of normal hatched larva and undeveloped embryos of *C. robusta* upon single exposure to non-functionalized PS NPs and BPA in FSW.

PS NPs and BPA Alone	% Normal	% Undeveloped
CTRL	75.43 ± 8.98	10.77 ± 8.88
CTRL (0.01% DMSO)	71.57 ± 5.17	10.92 ± 1.09
PS NPs 0.1 µg/mL	62.54 ± 10.10	21.81 ± 3.17
PS NPs 1 µg/mL	60.39 ± 8.73	23.63 ± 2.92
BPA 4.5 µM	11.18 ± 5.23	17.15 ± 1.81
BPA 10 µM	0	35.16 ± 10.82

**Table 3 biomolecules-12-01661-t003:** The percentage (%) of normal hatched larva and undeveloped embryos of *C. robusta* upon combined exposure without pre-incubation (T0) to non-functionalized PS NPs and BPA in FSW.

Combined (T0)	% Normal	% Undeveloped
CTRL	75.43 ± 8.98	10.77 ± 8.88
CTRL (0.01% DMSO)	71.57 ± 5.17	10.92 ± 1.09
PS NPs 0.1 µg/mL + BPA 4.5 µM	4.06 ± 3.54	18.04 ± 4.65
PS NPs 0.1 µg/mL + BPA 10 µM	0	26.1 ± 2.67
PS NPs 1 µg/mL + BPA 4.5 µM	0	25.77 ± 5.27
PS NPs 1 µg/mL + BPA 10 µM	0	55.47 ± 16.45

**Table 4 biomolecules-12-01661-t004:** The percentage (%) of *C. robusta* normally hatched larvae and undeveloped embryos upon combined exposure after 24 h incubation in FSW (T24) to non-functionalized PS NPs and BPA.

Combined (T24)	% Normal	% Undeveloped
CTRL	75.43 ± 8.98	10.77 ± 8.88
CTRL (0.01% DMSO)	71.57 ± 5.17	10.92 ± 1.09
PS NPs 0.1 µg/mL + BPA 4.5 µM	8.9 ± 1.81	26.56 ± 6.34
PS NPs 0.1 µg/mL + BPA 10 µM	0	78.39 ± 5.29
PS NPs 1 µg/mL + BPA 4.5 µM	11.1 ± 2.97	26.78 ± 6.00
PS NPs 1 µg/mL + BPA 10 µM	12.17 ± 1.2	26.54 ± 1.66

## Data Availability

Not applicable.

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
