# Peer review of "Short-Term Exposure to Nanoplastics Does Not Affect Bisphenol A Embryotoxicity to Marine Ascidian Ciona robusta"

_biomolecules, 2022, doi:10.3390/biom12111661_

Round 1
Reviewer 1 Report
Please find my comments and suggestions in the pdf document. In general, the manuscript needs to be improved to be possible its publication. The Results and Discussion sections are very confusing and not clear. The Conclusions section also needs improvements.

Reviewer 2 Report
Reviewer comments and suggestions
The authors in this study assessed the role of uncharged polystyrene nanoparticles (PS, 20 nm) as proxy for nanoplastics (1-10 µg/mL) alone and in combination with bisphenol A (4.5 and 10 µM) on Ciona robusta embryos by looking at embryotoxicity and phenotypic alterations. The study confirmed that the ability of BPA to affect ascidian C. robusta embryo development either alone and in combination with PS NPs. Additionally, the study reported no interactions are taking place between PS NPs and BPA in Filtered Natural Sea Water.
Overall the paper was good but in many places, they uses abbreviations without providing the full form at least the first time used in the manuscript. I suggest the authors should modify the manuscript in paragraphs so that readers could easily catch what the authors want to state in the manuscript.
A few concerns/comments needed to be explained/modified.
- Line 26 and 27 PS NPs and FSW first time used so it should be in full form in the abstract
- Line 44 [7-10] not 7,8,9,10
- Line 62 NSW (First time used)
- Line 69 “polystyrene nanoparticles” should be mentioned before as it was used several times
- Line 74 You can mention the harmful effects of BPA
- Typo error in line 81 and line 83 please delete A further and add “ The aim”
- Figure 1 BPA 4.5 and 10 uM, why there were three and two histograms. Please check the figure is this normal?
- First para of discussion “First para needs to be mentioning about the novelty of this study and then start with the next paragraph”
- Line 243, check the reference square brackets
- Line 250-251 Did the authors find any other studies which show interaction, that can also be added
- Please modify the Journal style of reference number 5, 29,31,32,35,39 and 42
Round 2
Reviewer 1 Report
The manuscript remains with serious flaws that must be corrected before publication be possible. The authors made improvements according to my suggestions/comments, however some issues show that the review was performed with little care/attention by the authors. Please consider the pdf document with my new comments. I highlighted (with yellow collour) in different parts of the manuscript some "errors" such as lack of spaces, etc. Furthermore, the abbreviations must be uniform throughout the manuscript, which is not the case (e.g. for the abbreviation of PS NPls).
The graphical abstract must not contains abbreviations.
Please review carefully again the manuscript in order to be able to publish it.
Thank you!
